# Faecal Carriage of Carbapenem-Resistant *Acinetobacter baumannii*: Comparison to Clinical Isolates from the Same Period (2017–2019)

**DOI:** 10.3390/pathogens11091003

**Published:** 2022-09-02

**Authors:** Bence Balázs, Zoltán Tóth, József Bálint Nagy, László Majoros, Ákos Tóth, Gábor Kardos

**Affiliations:** 1Department of Medical Microbiology, Faculty of Medicine, University of Debrecen, 4032 Debrecen, Hungary; 2Doctoral School of Pharmaceutical Sciences, University of Debrecen, 4032 Debrecen, Hungary; 3National Public Health Centre, 1097 Budapest, Hungary; 4Institute of Metagenomics, University of Debrecen, 4032 Debrecen, Hungary

**Keywords:** asymptomatic carriage, resistance reservoir, microbiota, class D carbapenemase

## Abstract

Increasing prevalence of *A. baumannii* was found in the faecal samples of inpatients without infection caused by *A. baumannii* (0.15%; 55/7806). The aim of the study was to determine whether there is a relationship between the clinical strains and the increased faecal occurrence. Characteristics of faecal and clinical isolates were compared between 2017 and 2019, and the direction of causality was assessed by Granger causality tests. In the case of the antibiotic resistance, faecal carriage of carbapenem-resistant *Acinetobacter baumannii* (CRAb) was Granger-caused by prevalence of CRAb in inpatients (F = 15.84, *p* < 0.001), but inpatient prevalence was not Granger-caused by CRAb faecal carriage (F = 0.03, *p* = 0.855). Whole genomes of 16 faecal isolates were sequenced by Illumina MiSeq; cgMLST types were determined. In faecal isolates, the occurrence of carbapenem resistance was lower than among the clinical isolates from the same period; only *bla_OXA-72_* harbouring ST636 and ST492 were detected, and the *bla_OXA-23_* harbouring ST2 and ST49 strains previously dominant in clinical isolates were absent. Carriage of *bla_OXA-72_* was linked to pMAL-1-like and pA105-2-like plasmids in ST636 and ST492 isolates, respectively, both in clinical and faecal isolates. The new ST636 and ST492 strains may colonise the gut microbiota of the patients, which thus may play a role as a reservoir.

## 1. Introduction

Healthcare-associated infections (HCAI) are among the most important emerging threats worldwide [1,2,3]. It is estimated that the number of deaths associated with multidrug-resistant (MDR) pathogens could reach 10 million by 2050, and the cost for control of resistance can reach USD 2.9 billion in the United States [1,4,5]. Moreover, the HCAIs caused by MDR bacteria are a major burden on the health care system; for example, ventilator-associated pneumonia lengthened the hospital stay by 9.1 and 38.7 bed-days in adult and neonatal intensive-care units (ICUs), respectively [6]. One of the major sources of HCAIs is the patient’s own microbiota; the intestinal microbiome is considered an important reservoir for these drug-resistant microorganisms and plays a crucial role not solely in the spread of antimicrobial-resistant strains but also acts as a hidden reservoir for genes conferring antibiotic resistance [7,8,9]. The rate of asymptomatic colonization of the gastrointestinal tract by extended-spectrum β-lactamase- (ESBL-) producing bacteria can reach 14% globally [10,11,12,13]; The colonisation with carbapenem-resistant *Enterobacterales* occurs less frequently (<10%) than with ESBL producers but shows an increasing rate [13].

Epidemiological studies indicate that MDR *A. baumannii* carriage rates are significantly lower compared to that of MDR *Enterobacterales* or enterococci as the latter are often part of the normal microbiota of healthy individuals, while *A. baumannii* is mainly environmental and rarely found in the intestinal tract of persons without prior hospital admission. However, intestinal colonisation, when present, may serve as an infectious source for the colonised or other patients. As carbapenem-resistant *A. baumannii* (CRAb) is a major agent in HCAI, faecal carriage of CRAb and its role in the strain dynamics of CRAb are worthy of attention [14].

Our working group has been following up the molecular epidemiology of CRAb clinical isolates since 2010 in the university [15,16]; the (ST2; ST49) dominant in the first part of the study period were replaced by *bla_OXA-72_* producers (ST636, ST492) in 2016–2017. The present study investigates the prevalence and antibiotic resistance of the faecal *A. baumannii* isolates collected between January 2017 and April 2019 in order to characterise the role of faecal CRAb carriage in their molecular epidemiology in comparison with contemporary clinical isolates from 2017.

## 2. Results

### 2.1. Prevalence, Susceptibility Testing and Resistance Genes

In an earlier study period (2011–2013) of faecal carriage only one *A. baumannii* isolate was found in 5862 faecal samples (0.02%), while between January 2017 and April 2019, 7806 faecal samples were investigated, and 55 *A.*
*baumannii* isolates (0.15%) were found in the faecal samples of inpatients, which is an order of magnitude higher than detected earlier. Other samples available from these carriers were negative for *A. baumannii*. Out of these 55 isolates, 30 originated from paediatric, 15 from internal medicine and 10 isolates from other wards; 15 of these 55 were resistant to imipenem and meropenem (CRAb), 17 to ciprofloxacin, 9 to amikacin and tobramycin and 13 to gentamicin. (Figure 1). None of the CRAb isolates originated from the faecal samples of paediatric patients. The *bla_OXA-40-like_* carbapenemase was found in 19 isolates, curiously, four carbapenem-susceptible isolates harboured a *bla_OXA-40-like_* gene. The earlier frequent *bla_OXA-23-like_* carbapenemases were absent, except for two isolates co-carrying *bla_OXA-23-like_* together with a *bla_OXA-40-like_* genes.

### 2.2. Whole Genome Sequencing (WGS)

Among the sequenced 16 faecal isolates, only *bla_OXA-72_* harbouring ST636 (n = 11) and ST492 (n = 2) were detected; two ST636 isolates carried *bla_OXA-72_* and *bla_OXA-23_* simultaneously (Table 1). Further investigation of the environment of the *bla_OXA-72_* gene revealed that the ST636 and ST492 isolates carry pMAL-1-like and pA105-2-like plasmids, respectively. Only three isolates belonged to other STs; the *aac(6′)-Ib-cr* and *bla_OXA-120_* (*bla_OXA-51-like_*) carrier ST132, the *bla_OXA-20_* (*bla_OXA-51-like_*) carrier ST45 isolate and the *bla_OXA-106_* (*bla_OXA-51-like_*) carrier isolate belonging to a novel sequence type (Table 1.). The WGS confirmed that both faecal and clinical ST636 isolates are genetically very close to each other, with only <= 4 alleles distance detected (Figure 2).

### 2.3. Comparison of Faecal and Clinical Isolates

The resistance rate among clinical isolates from 2017 were significantly higher to imipenem (95.4% vs. 27.3%; *p* < 0.001), meropenem (95.4% vs. 27.3%; *p* < 0.001), ciprofloxacin (96.9% vs. 30.9%; *p* < 0.001) and gentamicin (96.9% vs. 23.6%; *p* < 0.001) (Figure 1). This difference was not significant in the case of amikacin and tobramycin, but the resistance rates to these antibiotics were also higher in clinical isolates (Figure 1).

Prevalence of the *bla_OXA-40-like_* carbapenemase genes were significantly higher in clinical *A. baumannii* isolates (76.9% vs. 36.4; *p* < 0.001), two isolates carried only the *bla_OXA-23-like_* gene, while in the faecal isolates there were no strains carrying only the *bla_OXA-23-like_* carbapenemase. The proportion of isolates carrying two carbapenemases was also significantly higher among clinical isolates (15.4% vs. 3.6%; *p* < 0.05) (Figure 3). There was no significant difference for the aminoglycoside resistance *armA* gene (9.2% vs. 3.6%) (Figure 3).

The minimum inhibitory concentrations (MICs) of imipenem and meropenem were uniformly >32 mg/L in case of CRAb isolates. The time-kill analysis showed that there is no difference between the faecal and clinical strains in growth dynamics in the presence of carbapenems (data not shown). Meropenem was bactericidal at 128 mg/L in the case of ST492 isolates, but a bactericidal effect was never found against ST636, neither against isolates carrying *bla_OXA-72_* nor against isolates with both *bla_OXA-72_* and *bla_OXA-23_*. Imipenem was bactericidal at 256 mg/L against all investigated isolates.

The *bla_OXA-72_* carriage was linked to pMAL-1-like and pa105-2-like plasmids in ST636 and to ST492 isolates both in clinical and faecal isolates, and in one clinical isolate we found a complete pMAL-1 plasmid (GeneBank ID: KX230793.1) as a single contig with 99.99% identity. The *bla_OXA-23_* gene was linked to *Tn2008* (GeneBank ID: LN877214.1) transposon in all cases.

Faecal carriage of CRAb was Granger-caused by prevalence of CRAb in inpatients (F = 15.84, *p* < 0.001), but inpatient prevalence was not Granger-caused by CRAb faecal carriage (F = 0.03, *p* = 0.855). In contrast, neither faecal carriage of carbapenem-susceptible *A. baumannii* was Granger-caused by prevalence of carbapenem-susceptible *A. baumannii* in inpatients (F = 2.15, *p* = 0.155) nor vice versa (F = 0.13, *p* = 0.726).

## 3. Discussion

The study was inspired by the observation that the prevalence of CRAb in faecal samples increased significantly in the study period 2017–1019 compared to 2011–2013 (0.02% vs. 0.15%, *p* < 0.05; unpublished data). CRAb isolates originated from asymptomatic carriers, inpatients who did not show signs of *A. baumannii* infection, and it was not detected in any of their other samples. In comparison to the clinical isolates, the in vitro resistance was lower in the faecal *A. baumannii* isolates than in the clinical isolates in the case of all investigated antibiotics. The results of the WGS showed that only the newly appeared STs (ST636, ST492) could be found among the faecal CRAb isolates.

Several studies report the gut colonisation by CRAb in infected ICU patients, and prevalence rates were usually between 4.8% to 18.3%, but according to Corbella et al. (1996), the colonisation may reach 72.2%, and notably, a relationship was found between colonisation and nosocomial infections in ICUs, increased mortality rate and longer hospitalisation [17,18]. Our results are in concordance with the findings of Dijkshoorn et al., who reported 0.8% and 1.0% prevalence of *A. baumannii* in healthy individuals [19]. However, data is scant on the occurrence of CRAb among faecal samples of patients not infected by CRAb. Li et al. [20] reported a comparable faecal prevalence of 1.48% (74/5000) in an active surveillance of hospitalised patients using meropenem-containing selective medium, but in this report, it is unclear whether these had infection with *A. baumannii* or were colonised asymptomatically. The distribution of carbapenemase genes was markedly different between isolates from infectious sites and faecal isolates [20]. Eight, five, two and two *A. baumannii* isolates out of the sixteen were positive for *bla_OXA-23-like_*, *bla_OXA-40-like_*, *bla_VIM_* and *bla_NDM_* genes, respectively, though lack of the *bla_OXA-51-like_* gene in three isolates raises the possibility of misidentification in these cases [20].

In our setting, not only the incidence of faecal carriage of *A. baumannii* increased, but CRAb strains also appeared in the faeces of asymptomatic adult, but not paediatric, carriers. Faecal colonisation by CRAb, but not by carbapenem-susceptible *A. baumannii*, seems to be consequent to the prevalence of CRAb infections as indicated by Granger causality analysis. Similarly, colonisation of patients with ESBL-producing bacteria was shown to be consequent to the prevalence of infections earlier in this setting [21]. Accordingly, colonisation of the faecal microbiota seems to occur during hospital stay, which then becomes a risk factor for colonising healthcare personnel, roommates or future room occupants [22].

Furthermore, only the sequence types ST636 and ST492 harbouring *bla_OXA-72_* carbapenemase gene were found, while the previously dominant ST2 isolates carrying *bla_OXA-23_* were absent. This suggests that colonisation ability may be strain-specific, which, besides their higher resistance against meropenem [16], may have contributed to the epidemic success of *bla_OXA-72_* carrying strains [23,24,25], as their worldwide emergence seems to be contemporary to that of ST492 and ST636 [26].

Faecal carriage of multiresistant pathogens is an increasingly recognised problem, which is aggravated by lack of effective methods for eradication from colonised patients. Consequently, screening and isolation of positives patients remain most effective countermeasure against spread [27], necessitating increased infection control activities and allocation of time and resources.

The non-fermenting nosocomial Gram negatives *A. baumannii* and *P. aeruginosa* are generally accepted to persist mainly in abiotic reservoirs, such as water pipes or fomites, which is easier to contain using appropriate infection control practices [28,29]. Colonised individuals are thought to play only a minor or negligible role. As evidenced from the example of carbapenem-resistant *Enterobacterales* (CRE), faecal carriage may seriously hamper the eradication efforts of MDR pathogens from hospitals [30]. Therefore, appearance of CRAb strains with an increased ability to colonise the human gastrointestinal tract is alarming, since such a case will necessitate testing and infection control management of faecal carriage and not only the environmental contamination.

## 4. Materials and Methods

### 4.1. Isolates

Faecal samples of inpatients (n = 7806) sent for routine faecal culture between January 2017 and April 2019 at the Clinical Centre of the University of Debrecen were investigated for carriage of multiresistant Gram-negative bacteria. Samples were cultured on eosin-methylene blue agar (Neogen, Lansing, MI, USA) supplemented with 2 mg/L cefotaxime. After identification by matrix-assisted laser desorption/ionisation time-of-flight mass spectrometry (MALDI-TOF-MS) using a Microflex Biotyper (Bruker Daltonics, Billerica, MA, USA), *A. baumannii* isolates were investigated further. The susceptibility to amikacin, tobramycin, gentamicin, imipenem, meropenem and ciprofloxacin were determined by Kirby–Bauer method according to the European Committee on Antimicrobial Susceptibility Testing (EUCAST) guidelines (v 11.0) [31]; colistin susceptibility testing was performed by microdilution test (Merlin Diagnostics, Bornheim, Germany). Faecal isolates were compared to 65 clinical CRAb isolates (mainly from bronchial, wound and blood samples) collected in 2017 in our previous research [16].

The direction of causality between faecal carriage and isolations from clinical disease was analysed by Granger causality tests. Granger causality test is based on comparing forecast quality of one time-series alone and when including another time series. If the forecast quality is better in the latter case, then the first time-series is Granger caused by the other, indicating a potential causality [32]. We collected the monthly incidence of carbapenem susceptible *A. baumannii* and CRAb in faecal samples from inpatients and the monthly incidence densities of inpatients infected by carbapenem-susceptible *A. baumannii* and by CRAb per 100 occupied bed-days between 2017 and 2019. The Granger causality was tested between the respective pairs of time series [33]. Difference in susceptibility and resistance genes prevalence was analysed by chi-square test. The difference in imipenem and meropenem resistance between the clinical and faecal *A. baumannii* isolates was investigated by time-kill analysis [16].

### 4.2. Resistance Genes

Based on previous experience, carbapenemases *bla_OXA-23-like_*, *bla_OXA-24/40-like_*, *bla_OXA-51-like_* and the *armA* aminoglycoside resistance gene were sought by polymerase chain reaction (PCR), as described earlier [16,34,35].

### 4.3. Whole Genome Sequencing

Based on resistance gene carriage, representative CRAb (n = 13) as well as carbapenem-susceptible (n = 3) isolates were chosen for whole genome sequencing (WGS). The primary consideration for the selection criteria was the resistance gene profile, previously determined by PCR, and we also took into account phenotypic resistance. For isolates with the same resistance profile (genotype and phenotype), the origin of the faecal samples (hospital department) served as an additional criterion. Libraries were prepared with the Nextera DNA Flex library preparation kit (Illumina, San Diego, CA, USA) and sequenced with the MiSeq Reagent Kit v2 (300 cycles, Illumina) on the MiSeq platform (Illumina). The resulting FASTQ files were quality-trimmed before being de novo assembled with the Velvet assembler included in the Ridom SeqSphere+ software (Ridom GmbH, Münster, Germany). The cgMLST analysis was carried out using the Ridom SeqSphere+ software based on the ‘*A. baumannii* cgMLST’ version 1.0 scheme. The generated raw sequence reads uploaded to NCBI BioProject database (BioProject ID: PRJNA671692). The sequenced faecal isolates were compared to clinical isolates from our previous study (BioProject ID: PRJNA671692). The accession numbers of the faecal isolates are shown in Table 1. The antibiotic resistance genes were sought for in the genomes by ResFinder (v3.9; 90% ID threshold, 60% minimum length) and Comprehensive Antibiotic Resistance Database (CARD, v3.1.4).

## Figures and Tables

**Figure 1 pathogens-11-01003-f001:**
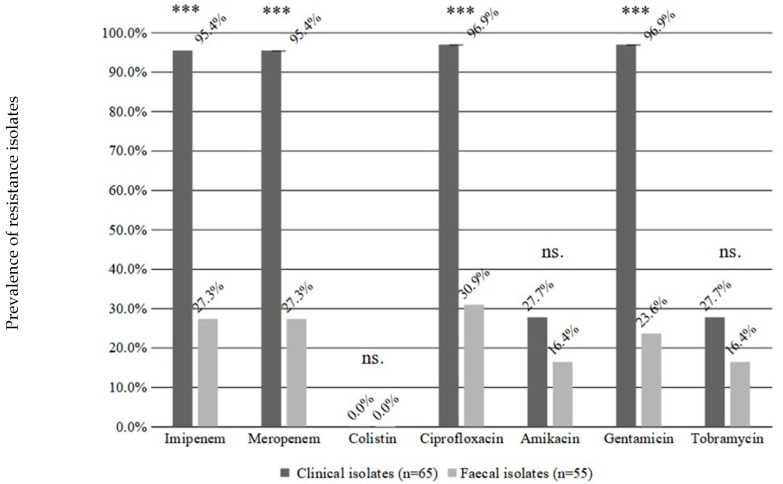
Comparison of resistance phenotypes between faecal and clinical isolates for different antibiotics. The percentages show the prevalence of resistant isolates (Y axis). Significance levels: *** *p* < 0.001; ns. not significant. Result of the clinical isolates from our previous work [16].

**Figure 2 pathogens-11-01003-f002:**
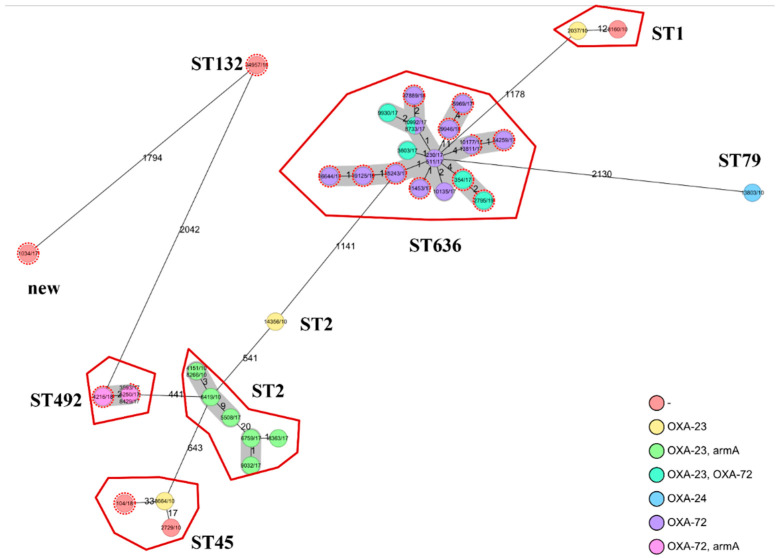
Minimum spanning tree based on cgMLST allelic profiles of A. baumannii isolates. Each circle represents an allelic profile based on sequence analysis of 2390 cgMLST target genes. Circles with dashed lines represent the faecal isolates, while the solid-contoured circles show the clinical isolates, colour-grouped by genotype. The numbers on the connecting lines illustrate the number of allelic differences. Closely related genotypes (<10 allelic differences based on presence-absence) are shaded. The cgMLST data of the clinical isolates were derived from our previous study [16].

**Figure 3 pathogens-11-01003-f003:**
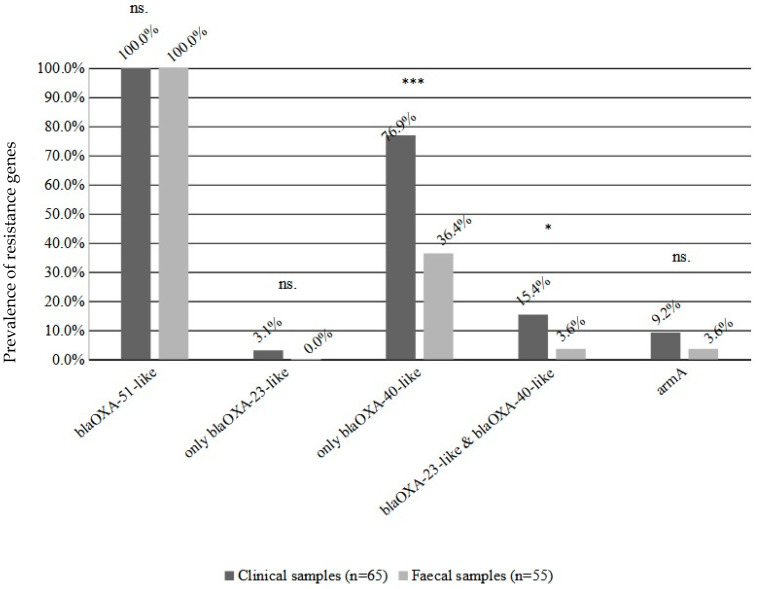
The prevalence of resistance genes among the clinical and faecal A. baumannii isolates. The percentages show the prevalence rate among the isolates (Y axis). Significance levels: * *p* < 0.05; *** *p* < 0.001; ns not-significant. The results of clinical isolates from our previous work [16].

**Table 1 pathogens-11-01003-t001:** The findings of whole genome sequencing antibiotic susceptibility testing of selected isolates and the relation between *Acinetobacter baumannii* sequence types and acquired carbapenemases.

Isolate	Year	ST	Acquired CHDLs	Resistance Phenotype						
				IMP	MEM	COL	CIP	AMN	GMN	TMN
ab61 Ac.No.: SRX14056986	2017	new	None	S	S	S	R	S	S	S
ab64 Ac.No.: SRX14056995	2017	492	blaOXA-72	R	R	S	R	R	R	R
ab65 Ac.No.: SRX14056996	2017	636	blaOXA-72	R	R	S	R	S	R	S
ab66 Ac. No.: SRX14056997	2017	636	blaOXA-72	S	S	S	R	R	S	R
ab67 Ac.No.: SRX14056998	2017	636	blaOXA-72	R	R	S	R	S	R	S
ab68 Ac.No.: SRX14056999	2017	636	blaOXA-72	R	R	S	R	S	S	S
ab69 Ac.No.: SRX14057000	2017	636	blaOXA-72	R	R	S	S	S	S	S
ab71 Ac.No.: SRX14056988	2017	636	blaOXA-72	R	R	S	R	S	R	S
ab72 Ac.No.: SRX14056989	2017	636	blaOXA-23 blaOXA-72	R	R	S	R	R	R	R
ab73 Ac.No.: SRX14056990	2017	636	blaOXA-72	R	R	S	R	S	R	S
ab74 Ac.No.: SRX14056991	2017	636	blaOXA-72	R	R	S	R	R	R	R
ab75 Ac.No.: SRX14056992	2017	636	blaOXA-72	R	R	S	R	S	R	S
ab60 Ac.No.: SRX14056985	2018	45	None	S	S	S	R	R	S	R
ab63 Ac.No.: SRX14056994	2018	492	blaOXA-72	R	R	S	R	R	R	R
ab70 Ac.No.: SRX14056987	2018	132	None	S	S	S	S	S	S	S
ab62 Ac.No.: SRX14056993	2019	636	blaOXA-23 blaOXA-72	R	R	S	R	R	R	R

IMP = imipenem; MEM = meropenem; COL = colistin; CIP = ciprofloxacin; AMN = amikacin; GMN = gentamicin; TMN = tobramycin; R = resistant; S = susceptible Ac.No. = accession number; ST = sequence type; CHDLs = carbapenem hydrolysing class D β-lactamase.

## Data Availability

The sequences of the bacterial strains were uploaded to the National Center for Biotechnology Information (NCBI) BioProject and BioSample databases. BioProject ID: PRJNA671692.

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
