# Peer review of "Faecal Carriage of Carbapenem-Resistant Acinetobacter baumannii: Comparison to Clinical Isolates from the Same Period (2017–2019)"

_pathogens, 2022, doi:10.3390/pathogens11091003_

Round 1
Reviewer 1 Report
General comments
The main aim of the manuscript appears to be to compare clinical to fecal A baumannii isolates and to assess causality between the two. However, important information is lacking from the manuscript (e.g. patient characteristics, timing of isolation relative to admission, prior exposures, follow-up data for patients with fecal carriage- e.g. did a CRAB infection develop later?). Furthermore, only fecal carriage is assessed and other potential sites of colonization are not evaluated. Finally, I have concerns regarding the methodology and rationale for assessing causality between fecal and clinical isolates.
Please find my comments/suggestions for each section below:
Abstract
- The abstract is a bit hard to follow and aims of the study are unclear.
Introduction
- I would remove the 1st paragraph (or significantly shorten to 1 sentence and merge with the 2nd paragraph)
Methods
- Please describe the routine practice in the hospital for fecal carriage screening (e.g. when is fecal carriage performed? at admission? every week during admission? are all admissions screened? etc)
- "Faecal isolates were compared to 65 clinical CRAb isolates from our previous research" Please clarify in Methods the time period during which these clinical isolates were collected (if I understand correctly based on the provided reference all clinical CRAB isolates were from 2017)
- "The direction of causality between faecal carriage (monthly prevalence in all faecal samples from inpatients) and isolations from clinical disease (infected inpatients per 100 occupied bed-days) was analysed by Granger causality tests". (a) All clinical CRAB isolates were from 2017 (based on ref 16), but fecal isolates were from 2017-2019. I don't understand how this analysis was conducted to assess causality between clinical (2017) and fecal (2017-2019) CRAB isolates. (b) If I understand correctly different sequence types were found in fecal (ST636; ST492) compared to clinical (ST2; ST49) strains . Based on this alone a correlation between fecal carriage and clinical isolates does not make sense
- I am not familiar with the Granger causality analysis. Perhaps a short discription in methods would be useful
Result
- I suggest adding a Table that describes the characteristics of patients who had A baumannii in fecal/clinical samples (e.g. age, ward of isolation, timing of isolation relative to admission)
Discussion
I suggest adding a paragraph in the begining to summarize the main findings of the study e.g. (1) increasing incidence of fecal carriage of CRAB, (2) different susceptibility pattern and sequence type comparing fecal to clinical AB etc
Author Response
Answer to comments of Reviewer 1:
We thank the Reviewer for the helpful comments and suggestions.
Comment 1: The abstract is a bit hard to follow and aims of the study are unclear.
Response: We replaced the text.
Comment 2: Introduction: I would remove the 1st paragraph (or significantly shorten to 1 sentence and merge with the 2nd paragraph)
Response: We shortened the first paragraph of the introduction, though we felt necessary to keep a few sentences on the key fac ts about the healthcare-associated infections and their burden.
Comment 3: - Please describe the routine practice in the hospital for fecal carriage screening (e.g. when is fecal carriage performed? at admission? every week during admission? are all admissions screened? etc)
Response: The faecal isolates were derived from our own screening for MDR pathogen carriage faecal samples sent for routine faecal culture for enteropathogens, as stated in the methods. Admission screening for faecal carriage was not in place in the hospital at the time of the sampling.
Comment 4: "Faecal isolates were compared to 65 clinical CRAb isolates from our previous research" Please clarify in Methods the time period during which these clinical isolates were collected (if I understand correctly based on the provided reference all clinical CRAB isolates were from 2017)
Response: We completed the text.
Comment 3: "The direction of causality between faecal carriage (monthly prevalence in all faecal samples from inpatients) and isolations from clinical disease (infected inpatients per 100 occupied bed-days) was analysed by Granger causality tests". (a) All clinical CRAB isolates were from 2017 (based on ref 16), but fecal isolates were from 2017-2019. I don't understand how this analysis was conducted to assess causality between clinical (2017) and fecal (2017-2019) CRAB isolates. (b) If I understand correctly different sequence types were found in fecal (ST636; ST492) compared to clinical (ST2; ST49) strains . Based on this alone a correlation between fecal carriage and clinical isolates does not make sense
Response A: Clinical CRAb isolates used for comparison were from 2017, but the Granger causality was based on the data between 2017 and 2019 both in case of faecal and in clinical isolates. We understood that this was not clear in the text; we amended this fault.
Response B: In our previous work, we investigated only clinical isolates. In that study, we found a strain replacement, the ST2 and ST49 strains encoded blaOXA-23 carbapenemase genes were dominant between 2010 and 2014, but in isolates collected 2016 and 2017 prevalence of the strains decreased significantly, in the same time new strains (ST636 and ST492) in our setting emerged, which carried blaOXA-72 carbapenemase, and some cases both mentioned resistance genes together. As a part of another investigation by our workgroup, we found increasing prevalence of A. baumannii in faecal samples, in parallel to the appearance of the new strains. In summary, among the clinical isolates we found the previously dominant STs and newly appeared STs, while in case of faecal CRAb isolates we can only found the new ones (see Figure 3).
Comment 4: I am not familiar with the Granger causality analysis. Perhaps a short discription in methods would be useful.
Response: A short description of the test was provided in the methods with a new reference added.
Comment 4: I suggest adding a Table that describes the characteristics of patients who had A baumannii in fecal/clinical samples (e.g. age, ward of isolation, timing of isolation relative to admission)
Response: We can not make any inclusion or exclusion criteria on sampling, the samples arrived to the laboratory as part of routine diagnostic, and we selected A. baumannii positive samples. Thus, we can not present these data due to GDPR and lack of permission of publishing patient data.
Comment 5: I suggest adding a paragraph in the begining to summarize the main findings of the study e.g. (1) increasing incidence of fecal carriage of CRAB, (2) different susceptibility pattern and sequence type comparing fecal to clinical AB etc
Response: We added the suggested statements to the first paragraph of the discussion.
Reviewer 2 Report
The authors have shown in a previous publication that high meropenem use has driven a shift in A. baumannii resistance genes from A. baumannii strains (ST2; ST49) producing blaOXA-23 during 2010-2014 to blaOXA-72-like carriers (ST636; ST492) during 2016-2017. In the present study, they investigate characteristics of faecal A. baumannii isolates (Ntotal=55, NCRAb=15) between January 2017 and April 2019. The blaOXA-40-like carbapenemase was found in 19 isolates, four carbapenem susceptible isolates harboured a blaOXA-40-like gene. Two isolates carried blaOXA-23-like together with blaOXA-40-like genes. Whole genome analysis was conducted in representative CRAb (n=13) as well as carbapenem susceptible (n=3) isolates. Faecal isolates were compared to 65 clinical CRAb isolates that were retrieved during 2017. Resistance rates and prevalence of the blaOXA-40-like carbapenemase genes were significantly higher in clinical A. baumannii isolates.
Comments
Line 10-11: “The dominant carbapenem resistant A. baumannii strains (ST2; ST49) producing blaOXA-23 were replaced by a blaOXA-72-like carriers (ST636; ST492) in 2014-2016”. Perhaps you mean during 2016-2017?
Line 32-34: “Most cases are caused by Enterococcus sp., Staphylococcus aureus, Clostridioides difficile, Acinetobacter baumannii, Pseudomonas aeruginosa and Enterobacterales, collectively termed as ESCAPE organisms” the correct term is ESKAPE K for K. pneumoniae and E for Enterobacter spp.
Line 58-60: “Present study investigates prevalence and antibiotic resistance of the faecal A. baumannii isolates, in order to characterise the role of faecal CRAb carriage in their molecular epidemiology in comparison with contemporary clinical isolates.” Please specify the time periods, ie between January 2017 and April 2019, that you investigated faecal iolates and the time period that the clinical strains were isolated, ie 2017.
Line 193: “Gram” instead of “Gramn”
Line 214: “Based on resistance gene carriage, representative CRAb (n=13) as well as carbapenem…” please specify the criteria for selecting these isolates for further analysis.
Line 184: “Carbapenem-resistant Enterobacteriaceae (CRE)” instead of “CRE”
Author Response
Answer to comments of Reviewer 2:
We thank the Reviewer for the helpful comments and suggestions.
Comment 1: Line 10-11: “The dominant carbapenem resistant A. baumannii strains (ST2; ST49) producing blaOXA-23 were replaced by a blaOXA-72-like carriers (ST636; ST492) in 2014-2016”. Perhaps you mean during 2016-2017?
Response: We have the gene occurrence data from 2010 to 2014 and then resumed the survey in 2016, finding that new strains (ST492, ST636) appeared and largely replaced the previous ones (ST2, ST49). Thus, we can only say somewhere between 2014 and 2016, as we have no gene prevalence data from 2015. However, the need for shortening the abstract eliminated this sentence.
Comment 2: Line 32-34: “Most cases are caused by Enterococcus sp., Staphylococcus aureus, Clostridioides difficile, Acinetobacter baumannii, Pseudomonas aeruginosa and Enterobacterales, collectively termed as ESCAPE organisms” the correct term is ESKAPE K for K. pneumoniae and E for Enterobacter spp.
Response: We modified the text as suggested.
Comment 3: Line 58-60: “Present study investigates prevalence and antibiotic resistance of the faecal A. baumannii isolates, in order to characterise the role of faecal CRAb carriage in their molecular epidemiology in comparison with contemporary clinical isolates.” Please specify the time periods, ie between January 2017 and April 2019, that you investigated faecal iolates and the time period that the clinical strains were isolated, ie 2017.
Response: We completed the text.
Comment 4: Line 193: “Gram” instead of “Gramn”
Response: We corrected the mistake.
Comment 5: Line 214: “Based on resistance gene carriage, representative CRAb (n=13) as well as carbapenem…” please specify the criteria for selecting these isolates for further analysis.
Response: We completed the text.
Comment 6: Line 184: “Carbapenem-resistant Enterobacteriaceae (CRE)” instead of “CRE”
Response: We corrected the text.
Reviewer 3 Report
The article “Faecal carriage of carbapenem resistant Acinetobacter baumannii: comparison to clinical isolates from the same period” investigates the presence carbapenem resistant Acinetobacter baumannii in faecal samples of inpatients and compared the antibiotic resistant profile and genes with clinical isolates. The study is interesting as it shows presence of carbapenem resistant Acinetobacter baumannii ST636 and ST492 in fecal samples and antibiotic resistant was more significant in clinical isolates as compared to faecal. However, the data generated from the whole genome has not been fully investigated to predict that the faecal strains can be reservoirs of infectious Acinetobacter baumannii strains
comments
1. Please mention the time period in the title
2. Line 10 please add an opening line indicating the problem statement
3. Line 14 Mention the total number of sample and % prevalence of CRAb
4. Line 24 Are ST636 and ST492 more virulent than ST2 and ST49
5. Line 63 please add these line in discussion where you compare your present results with earlier one
6. Line 135-137 it confusing please rephrase
7. Line # 191 what was the inclusion and exclusion criteria for inpatient fecal sampling
8. Line 201 please briefly explain the origin of clinical Isolates to make a fare comparison
910. Line 184 define CRE
111. Most of the results are not discussed in the discussion section like antibiotic resistant profiles why is it different in clinical and faecal samples and its implications, any reason why ST636 and ST492 have replaced ST2; ST49
112. It would be interesting to compare the virulence genes in clinical and fecal isolates to draw conclusion whether the fecal isolate have potential to cause infection or not.
Author Response
Answer to comments of Reviewer 3:
We thank the Reviewer for the helpful comments and suggestions.
Comment 1: Please mention the time period in the title
Response: We modified the title as suggested.
Comment 2: Line 10 please add an opening line indicating the problem statement
Response: We modified the abstract according to the suggestion of the other Reviewers, and now it starts with stating the observation initiating the study.
Comment 3: Line 14 Mention the total number of sample and % prevalence of CRAb
Response: We completed the text.
Comment 4: Line 24 Are ST636 and ST492 more virulent than ST2 and ST49
Comment 12: It would be interesting to compare the virulence genes in clinical and fecal isolates to draw conclusion whether the fecal isolate have potential to cause infection or not.
Response: There is no data on the difference between the virulence of different STs of A. baumannii to our best knowledge. The whole genomes determined (reference 16) did not reveal any systematic difference in the virulence gene composition between STs. We agree that this issue may worth further investigation. As faecal CRAb isolates was very close (0-4 allelic differences) it is conceivable that they may be able to cause disease. Replacement of previous STs by ST492 and ST636 may have been caused by their difference in meropenem susceptibility, as meropenem use was extensive during this period, as concluded in reference 16.
Comment 5: Line 63 please add these line in discussion where you compare your present results with earlier one
Response: The comparison was emphasized in the discussion.
Comment 6: Line 135-137 it confusing please rephrase
Response: We rephrased the text.
Comment 7: Line # 191 what was the inclusion and exclusion criteria for inpatient fecal sampling
Response: We can not make any inclusion or exclusion criteria on sampling, the samples arrived to the laboratory as part of routine diagnostic, and we selected A. baumannii positive samples. A sentence was added to show this.
Comment 8: Line 201 please briefly explain the origin of clinical Isolates to make a fare comparison
Response: We added the three most frequent sample types.
Comment 9: Line 184 define CRE
Response: Acronyms were defined.
Comment 10: Most of the results are not discussed in the discussion section like antibiotic resistant profiles why is it different in clinical and faecal samples and its implications, any reason why ST636 and ST492 have replaced ST2; ST49
Response: The question of strain replacement was thoroughly investigated earlier as reported in our previous study (reference 16), while the present study focuses on increased occurrence of faecal carriage and on the difference between the faecal and clinical isolates. Therefore, discussing the suggested issue would be redundant with the previous report (reference 16).
Round 2
Reviewer 1 Report
Authors have addressed all comments